# Perceived social support and associated factors among older people living in metropolitan cities of northwest Ethiopia: A community-based cross-sectional study

Habtamu Sewunet Mekonnen[1]*, Abere Woretaw Azagew[1], Chilot Kassa Mekonnen[1], Hailemichael Kindie Abate[1], Yohannes Mulu Ferede[1], Mohammed Hassen Salih[1], Nigusie Birhan Tebeje[2]

1 Department of Medical Nursing, School of Nursing, College of Medicine and Health Sciences, and Specialized Hospital, University of Gondar, Gondar, Ethiopia, 2 Department of Reproductive Health, Institute of Public Health, College of Medicine and Health Sciences, and Specialized Hospital, University of Gondar, Gondar, Ethiopia

* habtsew@ymail.com

**Data Availability Statement:** All relevant data are within the manuscript and its Supporting Information files.

## Abstract

### Background

Perceived social support is a complex construct that includes tangible and supportive feedback in addition to emotional, instrumental, appraisal, and informational support. Social support shields older adults from the negative effects of aging, such as illness and death, as well as the negative outcomes of stressful life events. The purpose of this study was to assess the perceived social support and associated factors among older people residing in metropolitan cities in northwest Ethiopia, as there is a dearth of evidence regarding this topic, particularly in the study area.

### Methods

A community-based cross-sectional study was carried out between December 19, 2020, and February 21, 2021. The systematic random sampling technique was used to select 830 study participants. Data collected using an interviewer administered questionnaire was entered using Epi-data version 4.6 and analyzed using Stata version 14. Both bivariate and multivariate logistic regression analyses were done. In the multivariate analysis, variables with P-value <0.05 were considered statistically significant. Adjusted odds ratio (AOR) with a 95% confidence interval was used to determine the strength and direction of the association.

### Results

A total of 816 participants were included with a 98.3% response rate. The mean age of participants was 68.2 (SD±7.2) years. In the current study, 339 (41.54%) of participants had low perceived social support. Having four and above children (AOR = 1.71, 95% CI = 1.10, 2.66), likely to have a severe mental illness (AOR = 0.33, 95% CI = 0.16, 0.68), medium and

**Funding:** The author(s) received no specific funding for this work.

**Competing interests:** The authors have declared that no competing interests exist.

**Abbreviations:** AOR, Adjusted odds ratio; CI, Confidence interval; FSSQ, Duke-UNC functional social support questionnaire; Katz ADL, Katz Index of Independence in Activities of Daily Living; K10, Kessler psychological distress scale; SD, Standard deviation; SOC, Sense of coherence.

high sense of coherence (AOR = 1.79, 95%CI = 1.05, 3.06) and 2.86 (AOR = 2.86, 95%CI = 1.44, 5.67), respectively, middle, fourth, and highest quantiles of wealth index (AOR = 1.97, 95%CI = 1.07, 3.60), 3.64 (AOR = 3.60, 95%CI = 1.91, 6.94), and 5.82 (AOR = 5.82, 95%CI = 2.90, 11.71),. Having no social service participation (AOR = 2.12, 95%CI = 1.06, 9.50) were significantly associated with low perceived social support.

## Conclusion and recommendations

In this study more than two-fifth of older people have low perceived social support. Number of children, mental health, sense of coherence, wealth index, and participation in social services were had a significant association with the perceived social support of older people. In order for these vulnerable populations to age healthily, alternative forms of support should be provided as the societal culture supporting the elderly has declined. Particularly, emphasis should be given for older people with mental health problems and those having no social service participation. Besides, further research is crucial targeting the actual social support of the elderly people including those living in the street, temporal residents, and religious places.

## Introduction

The term "social support" describes the material and psychological supports that a social network offers to people to help them deal with their problems [1]. The concept of perceived social support is multifaceted, encompassing emotional, instrumental, appraisal, and informational support, as well as supportive feedback and tangible support [2].

Older people are defined by the UN as those who are 60 years of age or older [3]. The official retirement age in Ethiopia is coincident with this definition, which has gained acceptance in that context [4]. Ageing is part of the life cycle that is an ongoing physiological cycle that eventually results in reduction in biological capacities and the body's capacity to adjust to metabolic stress. Additionally, aging is viewed as a complex phenomenon that involves social, psychological, and physical aspects [5]. Ageing has traditionally been associated with a decline in social, physical, and cognitive abilities [6].

Therefore, social support is crucial for maintaining relationships with others and one's general wellbeing as one age. At older life, people experience numerous changes that impact various aspects of their day-to-day interactions, including physical, psychosocial, economical, and spiritual wellbeing [7].

These can lead to reduced social interaction and, consequently, feelings of loneliness and isolation. The study among older people in Europe showed that receiving help from own children, relatives, neighbours /friends/, colleagues, and receiving home care were had significantly lower risk difference for depression, loneliness, nervousness, and sleep troubles [8]. Older people who receive social support are protected from the negative consequences of stressful life events like illness and death. Similarly, people who have someone to help them in later life are protected from the negative effects of aging [9].

Participants in a nine-year follow-up study of Alameda County residents showed a higher risk of death during the follow-up period when they had no social ties [10]. Higher levels of social support are associated with stronger immune systems and cardiovascular function,

according to an analysis of 81 studies. This result demonstrated the vital role that emotional support and family support systems play as a possible mechanism [11].

Perceived social support accounted for 11.7% of the variance in life satisfaction and 22.1% of the variance in life quality, according to a study done on the relationship between quality of life, satisfaction with life, and multidimensional perceived social support in older adults [12].

The study in Malaysia showed that the estimated mean of social support index score of older people was 27.65 and low monthly income, being single, no depression/normal, absence of activities of daily living, and dependency in instrumental activities of daily living were statically significant factors associated with the perceived social support [13].

Ethiopia is experiencing a sharp increase in life expectancy, which is contributing to the country's aging population [14]. In Ethiopian culture, older people were traditionally treated with great respect, given support by their families, and held in high regard even by outsiders. In the past, because family members provided them with all of their care and support, older people tended to stay in their own communities. Due to their advanced age, older adults who left their hometowns for any reasons could receive assistance and care wherever they went [15]. However, political unrest and internal strife have recently plagued Ethiopia, especially the Amhara regional state. These will have an impact on how well-supported older individuals are by their friends, neighbors, and community. In addition, elderly people will depart on their own without assistance as their children go to school and find employment elsewhere.

However, evidence about the Ethiopian older adults' perceived social support is scarce, despite the importance of the evidence and their being conducted over the world. Therefore, this study aimed to determine the prevalence of perceived social support and identify associated factors among older people living in metropolitan cities of northwest Ethiopia.

## Methods and materials

### Study design, period, and setting

A community-based cross-sectional study was carried out between December 19, 2020, and February 21, 2021. The methodological design guided and constructed using Exploratory Research Guide. The study was conducted in Gondar and Bahir Dar, two cities in northwest Ethiopia. There are an estimated 23348 older people in both cities who are 60 years of age or older.

Bahir Dar is the current capital city of the Amhara region in northern Ethiopia. It's a port on the south shore of the huge inland Lake Tana which is located 552 Km on driving distance from Addis Ababa (capital of Ethiopia). Administratively, Bahir Dar is a Special Zone and has 6 sub-cities and 3 Satellite towns. Gondar is a royal and ancient historical city of Ethiopia. It is found about 734 km driving distance away from Addis Ababa and 175.5 km from Bahir Dar. It has six sub-cities.

### Study population

Older people ($\geq$60 years old) who had lived for at least six months were the source population and all older people present during the specified data collection period were the study population. Older people who were residing in temporary settlements, streets, and temporary settlements were not included in the study.

### Sample size determination and sampling technique

"The sample size was determined using the single population proportion formula (n = $(Za/2)^2$ p (1-p)/$d^2$ (n = sample size needed; Z = confidence level at 95%; α = is the level of significance;

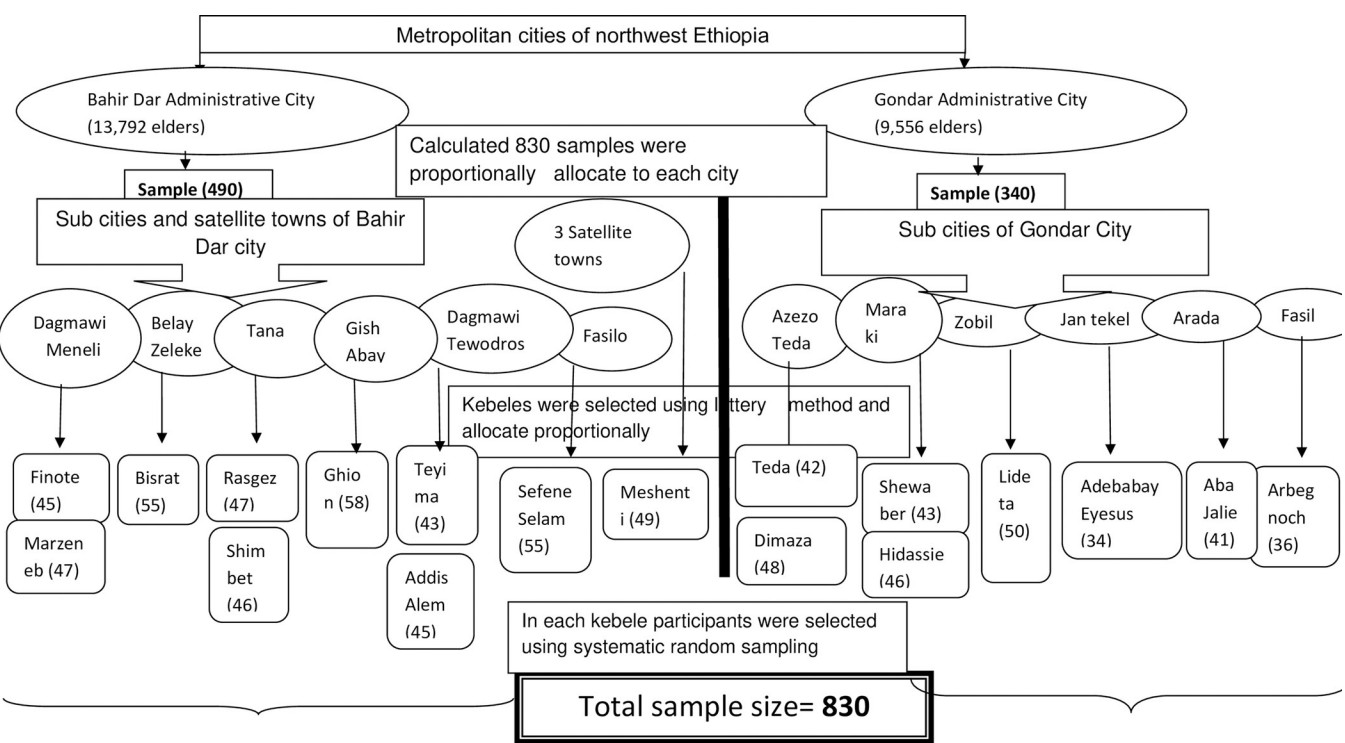

**Fig 1. Schematic presentation of the sampling procedure.**

p = Prevalence/proportion; and d = margin of error.) with the assumption of a 95% level of confidence, 5% marginal error, proportion (p) 56.9% taken from the pilot study, design effect 2 and 10% non-response rate. The final sample size was 830.

Each city was stratified into the sub-cities and in each sub-city; Kebeles (the lowest administrative level) were selected by the lottery method considering the number of Kebeles. In Bahir Dar administrative city, nine Kebeles and one satellite town were selected and, in Gondar administrative city, eight Kebeles were selected. Participants were allocated proportionally depending on the number of older people and were selected by systematic random sampling using the registered lists in each selected Kebele [16]." To choose the study participants, the sampling interval (K = 12) was computed by dividing the source population by the total sample size. The first participant was selected by lottery method. Fig 1.

### Data collection tool and procedures

"The data were collected using a face-to-face interview approach using a culturally adapted and validated structured questionnaire that was adapted by using an in-depth literature review. The questionnaire contains eight sections. The first section provides the socio-demographic characteristics of the study participants, the second section deals with social support, and the third is concerned with a health conditions, and the fourth section is concerned with nutritional, behavioral, and psychosocial characteristics. Section five to eight were about participation in various activities, mental health, sense of coherence, and urban wealth index-related questions, respectively. Communication with organizations that assist the elderly and influential individuals involved in issues of interest to the older people, such as health extension workers, took place during the data collection process. Then with the help of these people (workforce people), the participants were traced and interviewed in the quiet areas of their

homes after the interviewer briefly explained the purpose of the study and obtained consent from each participant. The data were collected and supervised by 18 trained BSc nurse data collectors and 9 MSc nurse trained supervisors [16]."

## Variables of the study

**Dependent variable.**

- Perceived Social Support

**Independent variables.**   The independent variables include.
*Socio-demographic variables.* Age, sex, marital status, level of education, religion, religious practice, occupation, economic status, presence of children, household size and living conditions.
*Health condition/status related variables.* Sense of coherence, self-rated health status, physical activity, functional ability, mental health, chronic disease, disability, health checkup
*Nutritional, risky behaviors, and psychosocial.* Frequency of meal, smoking, alcohol consumption, chat chewing, sedentary behavior, health insurance, self-perception, presence of a caregiver, participation in social activities.

## Operational definition

**Perceived social support.**   Social support is defined as the perceived availability of support, affection and instrumental aid from significant social partners, primarily family members and close friends [17], as well as neighbours and coworkers [18]. The perceived social support was assessed by using FSSQ. A score less than average was classified as indicating low perceived social support and a score equal to or greater than the average value was seen as indicating high perceived social support [19].

**Older people.**   "According to the UN definition, older persons are those people whose age is 60 years and above [3]. This definition has gained acceptance in the Ethiopian context as it coincides with the country's official retirement age [4]. Thus, in this study, people referred to as older people are all aged 60 years or older. Terminology for subgroups is as follows: Young-old 60–69, middle-old 70–79, and old-old > 80 years of age" [16, 20].

**Sense of coherence.**   "Sense of coherence was assessed by the sense of coherence scale SOC). Overall, scores 13–57, score 58–74, and score 75–91 were leveled as low, medium, and high sense of coherence" [16, 21].

**Participation in activities/physical activities.**   "This is the measure of personal activities, physical activities, activities with formal and informal support networks. The mean scores of the respondents on their levels of participation in the various activities were interpreted using the following scale: 1.00–1.80 = Very Low; 1.81–2.60 = Low; 2.61–3.40 = Moderate; 3.41–4.20 = High; and, 4.21–5.00 = Very High" [16, 22].

**Activities of daily living.**   "It was measured by the Katz ADL. The Index ranks adequacy of performance in the six functions of bathing, dressing, toileting, transferring, continence, and feeding. A score of 6 indicates the full function, 3–5 indicates moderate impairment, and 2 or less indicates severe functional impairment [16, 23]."

**Mental health.**   "Mental health was assessed using the K10. A score under 20 is likely to be well, score 20–24 likely to have a mild mental disorder, score 25–29 likely to have a moderate mental disorder, score 30, and over likely to have a severe mental disorder" [16, 24].

**Cigarettes smoking.** Those participants who were smoking during the interview were classified as Yes (current smoker) and those who smoked at least 100 cigarettes in his/her lifetime but not during the time of interview were classified as Yes (ever smoker) [25].

**Alcohol drinking.** Those participants who were drink alcohol at least 12 drinks in any one year in lifetime but no drinks in past year were classified as ever drinker (Yes) and those who drinks within the past one year classified current drinker (yes) [26, 27].

**Khat (chat) chewing.** Current khat chewing defined as khat chewing at least once weekly for the past one year. And those who were chewed khat in their life time classified as ever chewer (yes) [28].

**Known chronic disease.** The presence of chronic diseases were assessed by asking participants whether they have any chronic disease diagnosed by physicians or having follow-up and taken medication for the disease.

**Data quality control techniques.** "Culturally adapted and validated tools were used. Two days of training were given in each city for data collectors and supervisors to aid them in using the data collection tools and following the data collection procedures. A pilot trial of the questionnaire was carried out in the study area one week before the actual data collection. To ensure consistency of the collection technique and the acquisition of quality data, random checks were carried out by field supervisors and the principal investigator. Before the analysis, the collected data were checked for completeness and accuracy [16]."

**Data processing and analysis.** Before entering data, each questionnaire item was coded and checked for completeness. The data were entered using Epi-data version 4.6 and exported to Stata version 14 for data analysis. The data were presented using descriptive statistics, such as means, frequencies, and standard deviations. After conducting a bivariate logistic regression analysis, variables exhibiting P-values less than 0.25 were incorporated into the multivariate logistic regression for the final analysis. Hosmer and Lemeshow fitness of good test was computed. Variables with a P-value<0.05 were considered statistically significant and adjusted odds ratio with a 95% Confidence interval (CI) was computed to see the presence of strength and the direction of association between dependent and independent variables.

## Ethics approval and consent to participate

Ethical clearance was obtained from the institutional review board of the University of Gondar with the reference number V/P/RCS/05/2263/2020. Permission and supportive letters were secured from the respected cities and selected kebeles' administrative offices. Each study participant was informed about the purpose, method, expected benefit, and risk of the study. They also informed about their full right not to participate or withdraw from the study at any time and deciding not to participate had no impact on their services. Written informed consent was taken from study participants. For participants who were not read and write, a thumbprint was used in place of the participant's signature. Participants were guaranteed confidentiality and to ensure it, the information was identified using codes, and participants' names were not used.

## Results

### Socio-demographic characteristics of participants

Among 830 selected study participants, 816 were completely responded that comprises a 98.3% response rate. The participants mean age was 68.2 (SD±7.2) years. Of the participants, 433 (53.1%) were males and 511 (62.6%) were married. The majority, 766 (93.87%) had children, of which 369 (48.2%) were have 4–6 children. About 165 (20.2%) were in the middle quantile in the wealth index status. [Table 1].

**Table 1. Socio-demographic characteristics of elderly people in metropolitan cities of northwest Ethiopia, 2021 (n = 816).**

| Variables | n (%) | Variables | n (%) |
|---|---|---|---|
| Sex | | Religion | |
| Male | 433 (53.1) | Orthodox | 690 (84.6) |
| Female | 383 (46.9) | Muslim | 95 (11.6) |
| | | Protestant | 31 (3.8) |
| Age | | Religious practice | |
| Young—old | 548 (67.2) | Always | 466 (57.1) |
| Middle—old | 190 (23.3) | Sometimes | 188 (23.0) |
| Old-old | 78 (9.5) | Occasionally | 144 (17.7) |
| | | Never | 18 (2.2) |
| Marital status | | Current occupation | |
| Married | 511 (62.6) | Retired | 253 (31.0) |
| Widowed | 228 (27.9) | Employed | 72 (8.8) |
| Divorced | 77 (9.5) | Housewife | 153 (18.7) |
| | | Private work | 193 (23.7) |
| | | Non employed | 145 (17.8) |
| Educational Status | | Living condition | |
| Unable to read and write | 235 (28.8) | Live alone | 65 (8.0) |
| Able to read and write | 226 (27.7) | Live only with partner | 134 (16.4) |
| Grade 1–8 | 138 (16.9) | Live with children/grandchildren | 246 (30.1) |
| Grade 9–12 | 74 (9.1) | Live with partner/children/relatives | 371 (45.5) |
| Certificate and above | 143 (17.5) | | |
| Having children/live | | Wealth index | |
| Yes | 766 (93.87) | Lowest quantile | 164 (20.1) |
| No | 50 (6.13) | Second quantile | 164 (20.1) |
| | | Middle quantile | 165 (20.2) |
| | | Fourth quantile | 160 (19.6) |
| | | Highest quantile | 163 (20.0) |
| Number of live children (n = 766) | | | |
| 1–3 | 299 (39.0) | | |
| 4–6 | 369 (48.2) | | |
| >6 | 98 (12. 8) | | |
| Family size | | | |
| 1–3 | 208 (25.5) | | |
| 4–6 | 393 (48.2) | | |
| >6 | 215 (26.3) | | |

## Prevalence of perceived social support

In the current study, the prevalence of low perceived social support of older people was found to be 339 (41.54%) with 95% CI = 38.24, 44.32.

## Health condition of the study participants

Among participants, 471 (57.7%) were had good self-rated health status and 320 (39.2%) had chronic diseases. Of the participants, 738 (90.4%) had a fully function in their daily activities. and low levels of engagement in physical activity were reported by 328 (40.2%) participants. Likely to be well in mental health and medium sense of coherence were reported by 480 (58.8%), and 478 (58.6%), of the participants, respectively. [Table 2].

**Table 2. Health condition of the study participants in metropolitan cities of northwest Ethiopia, 2021 (n = 816).**

| Variables | Frequency (n) | Percent (%) |
|---|---|---|
| Self-rated health status | | |
| Good | 471 | 57.7 |
| Average | 242 | 29.7 |
| Bad | 103 | 12.6 |
| Known chronic disease | | |
| Yes | 320 | 39.2 |
| No | 496 | 60.8 |
| Physical disability | | |
| Yes | 60 | 7.6 |
| No | 756 | 92.6 |
| Health checkup | | |
| Yes | 383 | 46.9 |
| No | 433 | 53.1 |
| Daily living activities | | |
| Severe functional impairment | 33 | 4.1 |
| Moderate functional impairment | 45 | 5.5 |
| Full function | 738 | 90.4 |
| Participation in activities/physical activities | | |
| Very low | 165 | 20.2 |
| Low | 328 | 40.2 |
| Moderate | 249 | 30.5 |
| High | 55 | 6.8 |
| Very high | 19 | 2.3 |
| Mental health | | |
| likely to be well | 480 | 58.8 |
| likely to have a mild mental disorder | 156 | 19.1 |
| likely to have a moderate mental disorder | 84 | 10.3 |
| likely to have a severe mental disorder | 96 | 11.8 |
| Sense of coherence | | |
| Low | 159 | 19.5 |
| Medium | 478 | 58.6 |
| High | 179 | 21.9 |

## Nutritional, behavioral, and psychosocial characteristics of the study participants

477 (58.4%) of the participants in this study reported eating three times a day. Ever not smoke cigarettes and chewed khat were reported by 776 (95.1%) and 756 (92.6%) participants, respectively. Of the participants 495 (60.7%) were ever alcohol consumers. About 572 (70.1%) and 622 (76.2%) participants had good self-perception on aging life and had a caregiver, respectively. [Table 3].

## Factors associated with perceived social support of participants

The multivariate logistic regression analysis revealed that number of children, mental health, sense of coherence, wealth index, participation in social services were had a significant association with the perceived social support of older people.

The odds of low perceived social support of older people who had four and above children were 1.71 times (AOR = 1.71, 95% CI = 1.10, 2.66) higher than those who had less than four

**Table 3. Nutritional, behavioral, and psychosocial characteristics of the study participants in metropolitan cities of northwest Ethiopia, 2021 (n = 816).**

| Variables | Frequency | Percent |
|---|---|---|
| Meal frequency | | |
| Once per day | 17 | 2.1 |
| Two times per day | 243 | 29.8 |
| Three times per day | 477 | 58.4 |
| Four times per day | 79 | 9.7 |
| Living style | | |
| Have sedentary behavior | 57 | 7.0 |
| Sometimes do exercises/activities | 418 | 51.2 |
| Always do exercises/activities | 341 | 41.8 |
| Ever smoked cigarettes? | | |
| Yes | 40 | 4.9 |
| No | 776 | 95.1 |
| Currently, smoking cigarettes? | | |
| Yes | 15 | 1.8 |
| No | 801 | 98.2 |
| Ever chewed khat? | | |
| Yes | 60 | 7.4 |
| No | 756 | 92.6 |
| Currently, chewing khat? | | |
| Yes | 31 | 3.8 |
| No | 785 | 96.2 |
| Ever consumed any alcohol? | | |
| Yes | 495 | 60.7 |
| No | 321 | 39.3 |
| Alcohol consumption within the past 12 months? | | |
| Yes | 448 | 54.9 |
| No | 368 | 45.1 |
| Self-perception on aging life | | |
| Good | 572 | 70.1 |
| Somewhat good | 138 | 16.9 |
| Bad | 106 | 13.0 |
| Do you have a caregiver? | | |
| Yes | 622 | 76.2 |
| No | 194 | 23.7 |
| Health insurance | | |
| Yes | 277 | 34.0 |
| No | 539 | 66.0 |
| Participation in social services | | |
| Yes | 702 | 86.0 |
| No | 114 | 14.0 |

children. Compared to older adults who were likely to have a normal mental condition, those who were likely to have a severe mental illness were 67% (AOR = 0.33, 95%CI = 0.16, 0.68) less likely to report having low social support. The odds of low perceived social support of older people who had medium and high sense of coherence were 1.79 (AOR = 1.79, 95%CI = 1.05, 3.06) and 2.86 (AOR = 2.86, 95%CI = 1.44, 5.67), respectively, compared to those who had low sense of coherence. Older people who had middle quantile, fourth quantile, and highest

quantile of wealth index were had 1.97 (AOR = 1.97, 95%CI = 1.07, 3.60), 3.64 (AOR = 3.60, 95%CI = 1.91, 6.94), and 5.82 (AOR = 5.82, 95%CI = 2.90, 11.71) times higher low perceived social support, respectively, compared to those older people who were had low quantile wealth index. Compared to older adults who were no participated in social service to those who were participated in social services, the odds of low perceived social support were 2.12 (AOR = 2.12, 95%CI = 1.06, 9.50) times higher. [Table 4].

## Discussion

In the current study, the low perceived social support of older people was found to be 41.54% (95% CI = 38.24, 44.32).

The prevalence of low perceived social support of older people participated in the current study was higher than studies done in Spain [29] and India [30]. The study done among people aged 60 or over living in a community dwelling in Spain, in which only 35% of participants had low perceived social support. In India 8.04%, 38.47%, and 53.48% of the elderly were found to have low, moderate, and high social support, respectively. The difference might be due to the study time difference. The study in Spain was conducted in 2008. The time difference could result difference in the sociocultural interactions and support. The other possible reason for the discrepancy might be due to difference in study design and classification of the outcome variable. In Indian study, the design was observational cross-sectional study conducted for five months using MSPSS (Multi-dimensional Scale Perceived Social Support) questionnaire. In addition, the outcome variable was classified as low, moderate, and high social support.

Despite Ethiopia's long history of showing compassion, respect, and assistance for the older people, the current result indicated a decline in these attitudes and practices. This is supported with the Ethiopian economist reports which stated that while the country's overall rate of poverty is declining, the poorest people do not benefit from economic growth, and there is a high risk of shock-induced vulnerability, with older people being among the most severely affected members of society [31].

Ethiopia is currently dealing with a number of issues, such as political unrest, unemployment, drought, inflation, and food and housing insecurity for a large number of its elderly population. Two thirds of older adults in eastern Ethiopia who participated in a study on malnutrition were either malnourished or at risk of becoming malnourished [32]. These findings demonstrated how important it is to provide social support for vulnerable people, such as the older people. However, as the current study found, a significant portion of the older population perceives little social support. The finding is supported by the study conducted in Ethiopia, which shows that despite significant advancements in social and economic development over the past ten years, the country's spending on programs for older citizens' social protection has decreased [33].

Regarding the associated factors, number of children, mental health, sense of coherence, wealth index, and participation in social services were had a significant association with the perceived social support of older people.

The odds of low perceived social support of older people who had four and above children were 1.71 times (AOR = 1.71, 95% CI = 1.10, 2.66) higher than those who had less than four children. This could be because elderly parents with large families could receive different forms of support from their kids at different times. Children shared in the support duties as well, so their burden will lessen and enable them to give their parents the proper assistance. Consequently, support for these older individuals may not always come from others, including neighbors, friends, and the community. Hence, elderly individuals with a high child count

**Table 4. Multivariable logistic regression analysis of factors associated with perceived social support of older people in metropolitan cities of northwest Ethiopia, 2021 (n = 816).**

| Variables | Perceived Social Support (n) | | COR (95% CI) | AOR (95% CI) | p-value |
|---|---|---|---|---|---|
| | Low perceived social support | High perceived social support | | | |
| Sex | | | | | |
| Male | 157 | 276 | 1 | 1 | |
| Female | 182 | 201 | 0.63 (0.47, 0.83) | 0.95 (0.58, 1.56) | |
| Age | | | | | |
| Young-old | 207 | 341 | 1 | 1 | |
| Middle-old | 91 | 99 | 0.66 (0.47, 0.92) | 0.94 (0.60, 1.47) | |
| Old-old | 41 | 37 | 0.55 (0.34, 0.88) | 1.14 (0.55, 2.37) | |
| Marital Status | | | | | |
| Married | 184 | 327 | 1 | 1 | |
| Widowed | 114 | 114 | 0.56 (0.41, 0.77) | 1.48 (0.74, 2.94) | |
| Divorced | 41 | 36 | 0.49 (0.30, 0.80) | 1.31 (0.58, 2.95) | |
| Number of children | | | | | |
| 1–3 | 146 | 153 | 1 | 1 | |
| 4–6 | 112 | 257 | 2.19 (1.59, 3.01) | 1.71 (1.10, 2.66)* | **0.017** |
| >6 | 45 | 53 | 1.12 (0.71, 1.78) | 1.11 (0.58, 2.12) | 0.764 |
| Family size | | | | | |
| 1–3 | 120 | 88 | 1 | 1 | |
| 4–6 | 148 | 245 | 2.26 (1.60, 3.18) | 0.87 (0.53, 1.44) | |
| >6 | 71 | 144 | 2.77 (1.86, 4.12) | 0.86 (0.46, 1.63) | |
| Living condition | | | | | |
| Live alone | 50 | 15 | 1 | 1 | |
| Live only with partner | 59 | 75 | 4.24 (2.17, 8.28) | 1.90 (0.59, 6.17) | |
| Live with children/grandchildren | 114 | 132 | 3.86 (2.06, 7.24) | 2.62 (0.92, 7.43) | |
| Live with partner/children/relatives | 116 | 255 | 7.34 (3.95, 13.56) | 2.79 (0.90, 8.66) | |
| Current occupation | | | | | |
| Retired | 71 | 182 | 1 | 1 | |
| Employed | 31 | 41 | 0.52 (0.30, 0.87) | 0.74 (0.38, 1.45) | |
| Housewife | 68 | 85 | 0.49 (0.32, 0.74) | 0.93 (0.49, 1.76) | |
| Private work | 75 | 118 | 0.61 (0.41, 0.91) | 0.90 (0.53, 1.53) | |
| Non employed | 94 | 51 | 0.21 (0.14, 0.33) | 0.75 (0.39, 1.46) | |
| Known chronic disease | | | | | |
| Yes | 129 | 191 | 1 | 1 | |
| No | 210 | 286 | 0.92 (0.69, 1.22) | 0.76 (0.50, 1.15) | |
| Physical disability | | | | | |
| Yes | 40 | 299 | 1 | 1 | |
| No | 20 | 457 | 0.92 (0.69, 1.22) | 1.53 (0.71, 3.32) | |
| Mental health | | | | | |
| Likely to be well | 148 | 332 | 1 | 1 | |
| Likely to have a mild mental disorder | 75 | 81 | 0.48 (0.33, 0.70) | 0.69 (0.42, 1.10) | 0.121 |
| Likely to have a moderate mental disorder | 42 | 42 | 0.45 (0.28, 0.71) | 0.83 (0.44, 1.55) | 0.553 |
| Likely to have a severe mental disorder | 74 | 22 | 0.13 (0.80, 0.22) | 0.33 (0.16, 0.68)** | **0.003** |
| Daily living activities | | | | | |
| Severe functional impairment | 26 | 7 | 1 | 1 | |
| Moderate functional impairment | 28 | 17 | 2.26 (0.81, 6.31) | 1.67 (0.36, 7.69) | |
| Full function | 285 | 453 | 5.90 (2.53, 13.78) | 2.26 (0.55, 9.30) | |

*(Continued)*

**Table 4.** (Continued)

| Variables | Perceived Social Support (n) | | COR (95% CI) | AOR (95% CI) | p-value |
|---|---|---|---|---|---|
| | Low perceived social support | High perceived social support | | | |
| Participation in activities/physical activities | | | | | |
| Very low | 60 | 105 | 1 | 1 | |
| Low | 144 | 184 | 0.73 (0.50, 1.07) | 0.66 (0.40, 1.09) | |
| Moderate | 105 | 144 | 0.78 (0.52, 1.17) | 0.65 (0.38, 1.10) | |
| High | 22 | 33 | 0.86 (0.46, 1.60) | 0.97 (0.43, 2.21) | |
| Very high | 8 | 11 | 0.79 (0.30, 2.06) | 0.58 (0.18, 1.87) | |
| Living style | | | | | |
| Have sedentary behavior | 36 | 21 | 1 | 1 | |
| Sometimes do exercises/activities | 188 | 230 | 2.10 (1.18, 3.71) | 0.77 (0.30, 2.01) | |
| Always do exercises/activities | 115 | 226 | 3.37 (1.88, 6.04) | 1.34 (0.51, 3.57) | |
| Sense of coherence | | | | | |
| Low | 110 | 49 | 1 | 1 | |
| Medium | 192 | 286 | 3.34 (2.28, 4.90) | 1.79 (1.05, 3.06)* | **0.032** |
| High | 37 | 142 | 8.62 (5.26, 14.12) | 2.86 (1.44, 5.67)** | **0.003** |
| Meal frequency | | | | | |
| Once per day | 11 | 6 | 1 | 1 | |
| Two times per day | 121 | 122 | 1.85 (0.66, 5.16) | 0.87 (0.25, 3.01) | |
| Three times per day | 192 | 285 | 2.72 (0.99, 7.48) | 0.79 (0.23, 2.70) | |
| Four times per day | 15 | 64 | 7.82 (2.50, 24.52) | 3.79 (0.91, 15.76) | |
| Wealth index | | | | | |
| Lowest quantile | 125 | 39 | 1 | 1 | |
| Second quantile | 88 | 76 | 2.77 (1.73, 4.44) | 1.22 (0.68, 2.18) | 0.510 |
| Middle quantile | 60 | 105 | 5.61 (3.48, 9.06) | 1.97 (1.07, 3.60)* | **0.028** |
| Fourth quantile | 42 | 118 | 9.00 (5.44, 14.89) | 3.64 (1.91, 6.94)*** | **0.000** |
| Highest quantile | 24 | 139 | 18.56 (10.57, 32.59) | 5.82 (2.90, 11.71)*** | **0.000** |
| Health insurance | | | | | |
| Yes | 124 | 153 | 1 | 1 | |
| No | 215 | 324 | 1.22 (0.91, 1.64) | 0.99 (0.67, 1.45) | |
| Participation in social services | | | | | |
| Yes | 260 | 442 | 1 | 1 | |
| No | 79 | 35 | 0.26 (0.17, 0.40) | 2.12 (1.06, 9.50)* | **0.013** |

Note

\* Indicates the statistically significant variables with P-Value <0.05

\*\* Indicates the statistically significant variables with P-value < 0.01

\*\*\* Indicates the statistically significant variables with P-Value 0.000

may be perceived as having low social support from others. The finding is supported with the study done among older people with cancer in which patients with multiple children utilized more confidence support [34]. It is imperative to provide consideration and care for all older people, though, as the support that older people receive from their children is diminishing due to different factors. Similar study among Community-Dwelling Older Adults with Frailty and Pre-frailty in Hangzhou, China revealed that relationship with children was associated factors for the frail group [35].

Compared to older adults who were likely to have a normal mental condition, those who were likely to have a severe mental illness were 67% (AOR = 0.33, 95%CI = 0.16, 0.68) less

likely to report having low social support. This could be because, in cases where older people are likely to suffer from a serious mental illness, their friends, family, neighbors, and other community members help them in psychological, economical, and other supports. In addition, as many people considered the cause of mental illness related to demonic possession, evil spirit witchcraft and bewitchment, they would help them to access both modern and traditional medical care, as well as emotional supports and spiritual practices like receiving holy water. However, financial hardships and other issues will push all other older adults toward mental health illness if they don't receive the support they need. In the qualitative study about the "Holy water priest healers' views on collaboration with biomedical mental health services in Addis Ababa, Ethiopia", participants reported multiple explanations on the causes of mental illness—the most prevalent were in references to the Bible, demonic possession, evil spirit witchcraft and bewitchment and emphasized the need of collaboration between traditional healers and biomedical services, and potential ways to improve the condition of people with mental illness [36]. Similarly, the study conducted in Oslo showed that statistical association was observed between lack of social support and psychological distress [37].

The odds of low perceived social support of older people who had medium and high sense of coherence were 1.79 (AOR = 1.79, 95%CI = 1.05, 3.06) and 2.86 (AOR = 2.86, 95%CI = 1.44, 5.67), respectively, compared to those who had low sense of coherence. A sense of coherence has been defined as a global orientation toward life that involves cognitive, behavioral, and motivational elements and is expressed in the belief that the world is comprehensible, manageable, and meaningful [38]. Older people who possess a stronger sense of coherence may be able to handle their own problems and daily activities. In addition, if they possess a strong sense of coherence, they may think that their health and quality of life are good. So, they might not anticipate and require support from others. Consequently, it may be because of this reason that the older people with medium and high sense of coherence reported low perceived social support. This is supported with other study in which strong sense of coherence is related to good perceived health and quality of life [39]. This indicates that in order to reduce the number of people in need of support, it is critical to enhance older people's sense of coherence. The study among cervical cancer patients in China showed that sense of coherence was mediated the association between perceived social support and depressive symptoms [40].

Older people who had middle quantile, fourth quantile, and highest quantile of wealth index were had 1.97 (AOR = 1.97, 95%CI = 1.07, 3.60), 3.64 (AOR = 3.60, 95%CI = 1.91, 6.94), and 5.82 (AOR = 5.82, 95%CI = 2.90, 11.71) times higher low perceived social support, respectively, compared to those older people who were had low quantile wealth index. This could be because older people may be able to afford and obtain what they need if they are in the higher income quartiles. In addition, perceived income adequacy is positively associated with self-rated health and quality of life [41] that results low perceived social support. Similarly, the findings from the Malaysia National Health and Morbidity Survey 2018 revealed that income was associated with social support of older people [42].

Compared to older adults who were no participated in social service to those who were participated in social services, the odds of low perceived social support were 2.12 (AOR = 2.12, 95%CI = 1.06, 9.50) times higher. Individuals in Ethiopia engaged in social services such as eddir, ekub, mahiber, and other societal activities. These societal engagements are important for risk sharing, development of friendships, dispute resolution, haring and using timely information more effectively, lower level of funeral services anxiety, improvement of self-confidence and leadership role, reciprocity and coexistence and trust. In, addition, it is essential and distinctive in addressing emotional needs and sharing experiences; talk about their worries and urgent concerns [33]. Consequently, older individuals who did not take advantage of

these social services might not receive social support from others. In the study conducted in China, availability of recreational activities was associated with social support [35].

## Limitations

Due to the study's restriction to households in northwest Ethiopian metropolitan cities, it is possible that the sample does not adequately represent the older population residing in rural areas, temporary settlements, and streets. In addition, the older people's perceived social support was measured; however, actual support may differ from perceived support. In addition the cross-sectional study design cannot establish a cause-and-effect relationship. Since it is self-reported data and a face-to-face interviewer-administered technique, there is the possibility of social desirability bias and amnesia. However, to reduce these biases data collectors were well trained and participants were informed about the purpose of the study.

## Conclusion and recommendations

In this study more than two-fifth of older people have low perceived social support. Number of children, mental health, sense of coherence, wealth index, and participation in social services were had a significant association with the perceived social support of older people. In order for these vulnerable populations to age healthily, alternative forms of support should be provided as the societal culture supporting the elderly has declined. Particularly, emphasis should be given for older people with mental health problems and those having no social service participation. Besides, further research is crucial targeting the actual social support of the elderly people including those living in the street, temporal residents, and religious places.

## Supporting information

**S1 Data set.**
(DTA)

## Acknowledgments

The authors would like to thank the respective administration offices of the Bahir Dar and Gondar town for their permission letter and data collectors and supervisors for their commitment and the study participants for their valuable information.

## Author Contributions

**Conceptualization:** Habtamu Sewunet Mekonnen.

**Data curation:** Habtamu Sewunet Mekonnen.

**Formal analysis:** Habtamu Sewunet Mekonnen.

**Funding acquisition:** Habtamu Sewunet Mekonnen, Abere Woretaw Azagew.

**Investigation:** Habtamu Sewunet Mekonnen, Abere Woretaw Azagew, Nigusie Birhan Tebeje.

**Methodology:** Habtamu Sewunet Mekonnen, Abere Woretaw Azagew, Chilot Kassa Mekonnen, Hailemichael Kindie Abate, Yohannes Mulu Ferede, Mohammed Hassen Salih, Nigusie Birhan Tebeje.

**Project administration:** Habtamu Sewunet Mekonnen, Abere Woretaw Azagew, Chilot Kassa Mekonnen, Hailemichael Kindie Abate, Yohannes Mulu Ferede, Mohammed Hassen Salih, Nigusie Birhan Tebeje.

**Resources:** Habtamu Sewunet Mekonnen, Abere Woretaw Azagew, Chilot Kassa Mekonnen, Hailemichael Kindie Abate, Yohannes Mulu Ferede, Mohammed Hassen Salih, Nigusie Birhan Tebeje.

**Software:** Habtamu Sewunet Mekonnen, Abere Woretaw Azagew, Chilot Kassa Mekonnen, Hailemichael Kindie Abate, Yohannes Mulu Ferede, Mohammed Hassen Salih, Nigusie Birhan Tebeje.

**Supervision:** Habtamu Sewunet Mekonnen, Abere Woretaw Azagew, Chilot Kassa Mekonnen, Hailemichael Kindie Abate, Yohannes Mulu Ferede, Mohammed Hassen Salih, Nigusie Birhan Tebeje.

**Validation:** Habtamu Sewunet Mekonnen, Abere Woretaw Azagew, Chilot Kassa Mekonnen, Hailemichael Kindie Abate, Yohannes Mulu Ferede, Mohammed Hassen Salih, Nigusie Birhan Tebeje.

**Visualization:** Habtamu Sewunet Mekonnen, Abere Woretaw Azagew, Chilot Kassa Mekonnen, Hailemichael Kindie Abate, Yohannes Mulu Ferede, Mohammed Hassen Salih, Nigusie Birhan Tebeje.

**Writing – original draft:** Habtamu Sewunet Mekonnen.

**Writing – review & editing:** Habtamu Sewunet Mekonnen, Abere Woretaw Azagew, Chilot Kassa Mekonnen, Hailemichael Kindie Abate, Yohannes Mulu Ferede, Mohammed Hassen Salih, Nigusie Birhan Tebeje.

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
