## [Decision Letter · Decision Letter 0]

2 Oct 2024

PONE-D-24-24802Perceived social support and associated factors among older people living in metropolitan cities of northwest Ethiopia: A community-based cross-sectional studyPLOS ONE

Dear Dr. Mekonnen,

Thank you for submitting your manuscript to PLOS ONE. After careful consideration, we feel that it has merit but does not fully meet PLOS ONE’s publication criteria as it currently stands. Therefore, we invite you to submit a revised version of the manuscript that addresses the points raised during the review process.

We look forward to receiving your revised manuscript.

Kind regards,

Wudneh S Belay (PhD Student, Flinders University)

Academic Editor

PLOS ONE

Journal Requirements:

2. In your revision ensure you cite all your sources (including your own works), and quote or rephrase any duplicated text outside the methods section. Further consideration is dependent on these concerns being addressed.

“The authors would like to express our gratitude to the University of Gondar for the fund and the approval of the ethical clearance.”

4. In the online submission form you indicate that your data is not available for proprietary reasons and have provided a contact point for accessing this data. Please note that your current contact point is a co-author on this manuscript. According to our Data Policy, the contact point must not be an author on the manuscript and must be an institutional contact, ideally not an individual. Please revise your data statement to a non-author institutional point of contact, such as a data access or ethics committee, and send this to us via return email. Please also include contact information for the third party organization, and please include the full citation of where the data can be found.

Reviewers' comments:

Reviewer's Responses to Questions

**Comments to the Author**

1. Is the manuscript technically sound, and do the data support the conclusions?

Reviewer #1: Yes

Reviewer #2: Yes

2. Has the statistical analysis been performed appropriately and rigorously? 

Reviewer #1: Yes

Reviewer #2: Yes

3. Have the authors made all data underlying the findings in their manuscript fully available?

Reviewer #1: Yes

Reviewer #2: Yes

4. Is the manuscript presented in an intelligible fashion and written in standard English?

Reviewer #1: No

Reviewer #2: Yes

5. Review Comments to the Author

Reviewer #1: I applaud the author's motivation for putting this notion into action. Based on my rigorous review, I have some suggestions for improving the manuscript's quality and increasing its chances of approval. By expressing this, I intend to use both general and specific comments components appropriately.

General Comments

1. The article lacks continuous line numbers; why not put in line numbers? It is simple to provide comments and suggestions using point line numbers, and when addressing the offered remarks, remember to designate addressed comments with line numbers and page numbers.

2. What concerns me the most about this article are the tools used by the authors to assess the outcome variable perceived social support, and why you (authors) prefer FSSQ over more comprehensive tools used to assess perceived social support, such as the Multidimensional Scale of Perceived Social Support (MSPSS). FSSQ is typically used to assess the role of social support on medication adherence among patients with mental disorders, rather than general population perceived social support. Even it contains two components: confidant support and affective support, the scoring technique is continuous rather than dichotomous. How did you classify your outcome variable as low or high? This means that methods of scoring with suitable citation are required.

3. Why are you interested in this topic? What is the expected value provided by this work, given that perceived social support is a well-studied topic around the world, even among particular populations such as the elderly? So, in the introduction part, convey the peculiarity of this title boldly.

4. I came across a paper titled "Satisfaction with life and associated factors among elderly people living in two cities in northwest Ethiopia: a community-based cross-sectional study" while reading past research on perceived social support in Ethiopia and around the world. This paper is practically identical to the current paper, including the authors, sample size, study location, and study population, so could you please clarify this issue?

Specific Comments/component wise comments

Abstract

Please replace "the purpose of this study was to ascertain" with appropriate action verbs or describe how you would ascertain perceived social support among the study population in sentence four of the Background paragraph.

Methods

In the multivariate analysis, factors having a P-value ≤ 0.05 were considered statistically significant. Are there variables with a P-value of 0.05 that are deemed statistically significant? Although the scientific term is "p-value less than or equal to 0.05," most scholars prefer "p-value less than 0.05." That means, are there marginally significant variables?

Results

In sentence three, it says "likely to have a severe mental illness"??? It does not make sense, so change it to "participants having severe mental illness" or "an individual having severe mental illness."

Conclusion and recommendation

Your conclusion and recommendations should be based on your findings, which indicates that modifiable variables such as mental illness and lack of social participation should be addressed by stakeholders or future studies. illness"

Introduction

Change the word "background" to "introduction."

After writing an overview or explanation of social support, please explain what it means for older persons. That means add overview of older person.

“As a result, research on the extent of perceived social support and the factors that influence it is important and timely for older adults, the community, legislators, implementers, and other relevant organizations that work with older adults.” Please rephrase this sentence to make it a sense.

Methods and Materials

Study design, period, and setting.... Add periods between design and area or settings, and notify the reader what guidelines were used to guide the construction of the methodological design.

SAMPLE SIZE DETERMINATION AND SAMPLING TECHNIQUE

First, why did you choose to use proportions from a pilot study instead of previous studies in the same country or socioeconomic setting?

Second, on line three, it states "22 design effect 2". Kindly clarify it. Is Design Effect 2 or 22?

Third, use a formula editor, such as MathType, to clarify sample size calculation procedures for readers.

Fourth, you stated that study participants were recruited using systematic random sampling from registration lists. Could you please share or write the interval used to pick them?

Finally, a schematic presentation of the sampling technique for assessing perceived social support between the two cities and sub-cities should be included.

Operational Definition

The authors' choice of this instrument over the Multidimensional Scale of Perceived Social Support, which has superior psychometric qualities, remains unclear. I require compelling justification.

Operationalize what is “older people”, “Young – old”, “Middle – old”, and “Old-old” mean and use appropriate references.

Substance addiction is listed as an explanatory variable, which requires operationalization using appropriate tools. Define the terms "ever substance users" and "current substance users." Refer to ASSIST tools.

Results

Of the 830 participants, 816 were enrolled in the study. Explain why 14 were excluded. If there is missing ,explain way of management.

What is your benchmark for categorizing variables such as money, living conditions, and family size?

Table 1 has contradictory data, such as "Having children /live/…. Yes 764 (93.6)" and "Number of live children (n =766)".Please clarify whether this is correct or incorrect.

MAGNITUDE PREVALENCE OF PERCEIVED SOCIAL SUPPORT

Use the word prevalence instead of magnitude

When reporting the prevalence of perceived social support among older individuals, it is preferable to provide simply the prevalence (percentage) of low perceived social support with a 95% confidence interval because the prevalence of high perceived social support among these populations is already known.Because your stated low PSS.

HEALTH CONDITION OF THE STUDY PARTICIPANTS

Please explain whether the chronic medical and mental diseases listed in Table 2 were diagnosed by a physician, as well as what specific illnesses included in the chronic medical illness category.

NUTRITIONAL, BEHAVIORAL, AND PSYCHOSOCIAL CHARACTERISTICS OF THE STUDY PARTICIPANTS

As I have already gave you suggestion first define ever and currents substance use amend accordable, for example in Table 3 it says “Alcohol consumption within the past 12 months” ,there is no such assessment in the case of substance use, according to alcohol, smoking and substance involvement screening tool (ASSIST) tools using at least one of a specific substance (Alcohol, Khat, Cigarette, and other substances) for nonmedical purposes within last 3 months according to is regarded as current substance users and using at least one of any specific substance (Alcohol, Khat, Cigarette, and other substances) for nonmedical purposes at least once in a lifetime according to ASSIST refers ever substance users ,so correct according to this tools.

FACTORS ASSOCIATED WITH LIFE SATISFACTIONPERCIEVED SOCIAL SUPPORT OF ELDERLY PEOPLE

Is your outcome variable life satisfaction or perceived social support???

under the table 4 model fitness value should be added as footnotes.

Discussion

As I previously stated, it is preferable to focus on low perceived social support, and a 95% confidence interval is essential when comparing your findings to research conducted around the world.

In the second paragraph, is your study higher or lower than those of Spain and India? As far as I can see, your prevalence of low perceived social support is higher than studies conducted in both nations. So, please check and clarify for the reader. Include reasons for any discrepancies or differences in prevalence between your study and theirs. This includes the following: "In India 8.04%, 38.47%, and 53.48% of the elderly were found to have low, moderate, and high social support, respectively" …This category is based on the Oslo Social Support -3 items.

Regarding factors associated with low perceived social support,you merely interpreted the findings; please include comparisons with other studies conducted around the world.

Limitations

Add limitations to the study design as well.

Furthermore, because it is self-reported data and a face-to-face interviewer-administered technique, there is the possibility of social desirability bias and amnesia, so consider these limitations as well.

Conclusion and recommendations

It should be based on your findings, with a particular emphasis on modifiable factors identified as being connected with your outcome variable.

That’s it for now, and I am looking forward for reading your manuscript in its final form.

Reviewer #2: The authors have choosen an interesting topic. I have some issues that needs yours justification. Individuals who were living on the street and in religious institution, i.e homeless individuals were excluded in your study. Why? I think homeless individuals needs social support in all circumstance's. I like your topic if it includes this vulnerable population groups.

6. PLOS authors have the option to publish the peer review history of their article (what does this mean?). If published, this will include your full peer review and any attached files.

Reviewer #1: No

Reviewer #2: **Yes: **Agmas Wassie Abate

---

## [Author Response · Author response to Decision Letter 0]

12 Oct 2024

Authors’ point by point response to editor and reviewers

Dear reviewers and journal editor/s we would like to thank you for your concerns and important suggestions. Really all the comments and suggestions are crucial to improve the manuscript. We have made revision throughout the manuscript and this is a point by point response for your comments. 

Authors’ point by point response to editors 

Response: Thank you dear editor, we have provided our responses for each points and reviewer and labeled the letter as ’response to reviewers’

Response: We have uploaded the revised version labeled ‘Revised manuscript with Tack Changes’

Response: We have uploaded unmarked version labeled as ‘Manuscript’ 

Journal Requirements:

https://journals.plos.org/plosone/s/file?id=wjVg/PLOSOne_formatting_sample_main_body.pdf andhttps://journals.plos.org/plosone/s/file?id=ba62/PLOSOne_formatting_sample_title_authors_affiliations.pdf

Response: We have revised the manuscript and addressed the PLOS ONE’s style requirements. 

2. In your revision ensure you cite all your sources (including your own works), and quote or rephrase any duplicated text outside the methods section. Further consideration is dependent on these concerns being addressed

Response: Thank you, we cited the references, rewrite the texts, and in some of the texts we have used quote.

“The authors would like to express our gratitude to the University of Gondar for the fund and the approval of the ethical clearance.”

Response: Thank you we have deleted the funding information. Yes, the statement “The author(s) received no specific funding for this work.” is correct. 

4. In the online submission form you indicate that your data is not available for proprietary reasons and have provided a contact point for accessing this data. Please note that your current contact point is a co-author on this manuscript. According to our Data Policy, the contact point must not be an author on the manuscript and must be an institutional contact, ideally not an individual. Please revise your data statement to a non-author institutional point of contact, such as a data access or ethics committee, and send this to us via return email. Please also include contact information for the third party organization, and please include the full citation of where the data can be found.

Response: We have taken amendment in this regard. We will revise the online submission; the data set what we used for the analysis will be available.

Response: The ethics statement moved to the method section (Page 7; lines 134-142).

Authors’ point by point response to reviewer 1

General Comments

1. The article lacks continuous line numbers; why not put in line numbers? It is simple to provide comments and suggestions using point line numbers, and when addressing the offered remarks, remember to designate addressed comments with line numbers and page numbers.

Response: Yes, we agreed and put continuous line numbers. 

2. What concerns me the most about this article are the tools used by the authors to assess the outcome variable perceived social support, and why you (authors) prefer FSSQ over more comprehensive tools used to assess perceived social support, such as the Multidimensional Scale of Perceived Social Support (MSPSS). FSSQ is typically used to assess the role of social support on medication adherence among patients with mental disorders, rather than general population perceived social support. Even it contains two components: confidant support and affective support, the scoring technique is continuous rather than dichotomous. How did you classify your outcome variable as low or high? This means that methods of scoring with suitable citation are required.

Response: That is true. There are studies worldwide using different tools as the measurement of perceived social support. The Multidimensional Scale of Perceived Social Support (MSPSS) is one of the tools utilized by the researchers. However, 

- Even if the FSSQ is utilized to measure the perceived social support associated with medication adherence, the questions are measuring general supports rather specific supports to medications adherence. 

- MSPSS tool also consisting general questions related to the social support.

- MSPSS tool validated among the general population, but the FSSQ is validated to the older people. 

- Before the data collection we have been validated the FSSQ in the target older people (general older population)

- The FSSQ has fewer questions than MSPSS. So, since the current study population is older people, having fewer questions are suitable for such people. 

- There are also previous studies using the FSSQ.

- “Responses to each question are scored on a 1 to 5 scale. ”As much as I would like" receives a score of 5 and "Much less than I would like" receives a score of 1. The scores from all eight questions are summed (maximum 40) and then divided by 8 to get an average score. The higher the average score, the greater the perceived social support.” This is cited within the manuscript “reference 14”. http://adultmeducation.com/AssessmentTools_4.html

3. Why are you interested in this topic? What is the expected value provided by this work, given that perceived social support is a well-studied topic around the world, even among particular populations such as the elderly? So, in the introduction part, convey the peculiarity of this title boldly.

Response: Thank you we agreed and add statement regarding the issue (Page 6; lines 104-107)

4. I came across a paper titled "Satisfaction with life and associated factors among elderly people living in two cities in northwest Ethiopia: a community-based cross-sectional study" while reading past research on perceived social support in Ethiopia and around the world. This paper is practically identical to the current paper, including the authors, sample size, study location, and study population, so could you please clarify this issue?

Response: Yes, you are right. Thank you for reading our previous published work. We have large data with many objectives. Still we have data ready for analysis and the current study is one of the objectives taken from the large data set. 

Specific Comments/component wise comments

Abstract

Please replace "the purpose of this study was to ascertain" with appropriate action verbs or describe how you would ascertain perceived social support among the study population in sentence four of the Background paragraph.

Response: Thank you. We rephrased it as “assess” 

Methods

In the multivariate analysis, factors having a P-value ≤ 0.05 were considered statistically significant. Are there variables with a P-value of 0.05 that are deemed statistically significant? Although the scientific term is "p-value less than or equal to 0.05," most scholars prefer "p-value less than 0.05." That means, are there marginally significant variables?

Response: Thank you; there was no variable with P-value of 0.05. As it is indicated in the regression table, all the statistically significant variables were had P-value <0.05. We just write there considering the general agreement not the actual value in this study. To clear the confusion we make it < 0.05. 

Results

In sentence three, it says "likely to have a severe mental illness"??? It does not make sense, so change it to "participants having severe mental illness" or "an individual having severe mental illness."

Response: Really thank you for your great observation. However, this variable was measured by using Kessler psychological distress scale. This scale helps as a screening of the likelihood of having a mental disorder. It does not measure the actual mental disorder. "likely to have a severe mental illness” was written as the scale developers preference. The term “individual having severe mental illness” is good but it seems for the actual mental disorder. We hope these clarifications make the confusion clear. https://www.worksafe.qld.gov.au/data/assets/pdf_file/0010/22240/kessler-psychological-distress-scale-k101.pdf

Conclusion and recommendation 

Your conclusion and recommendations should be based on your findings, which indicates that modifiable variables such as mental illness and lack of social participation should be addressed by stakeholders or future studies. illness"

Response: thank you, we agreed and add a statement regarding the issue. Page 3; line 53-54

 Introduction

Change the word "background" to "introduction."

After writing an overview or explanation of social support, please explain what it means for older persons. That means add overview of older person.

“As a result, research on the extent of perceived social support and the factors that influence it is important and timely for older adults, the community, legislators, implementers, and other relevant organizations that work with older adults.” Please rephrase this sentence to make it a sense.

Response: We have addressed the concerns under the introduction section. 

Methods and Materials 

Study design, period, and setting.... Add periods between design and area or settings, and notify the reader what guidelines were used to guide the construction of the methodological design. 

Response: Thank you, we have done it (Page 6; lines 113-116) 

SAMPLE SIZE DETERMINATION AND SAMPLING TECHNIQUE

First, why did you choose to use proportions from a pilot study instead of previous studies in the same country or socioeconomic setting?

Second, on line three, it states "22 design effect 2". Kindly clarify it. Is Design Effect 2 or 22?

Third, use a formula editor, such as MathType, to clarify sample size calculation procedures for readers.

Fourth, you stated that study participants were recruited using systematic random sampling from registration lists. Could you please share or write the interval used to pick them?

Finally, a schematic presentation of the sampling technique for assessing perceived social support between the two cities and sub-cities should be included

Response: Thank you for the concerns.

For the first concern: The data for this specific manuscript were taken from the large data set. Since couldn’t find similar study among the older population in Ethiopia, we preferred to estimate the proportion from the pilot study rather considering proportion from other countries or taking 50%. Considering the pilot study is preferable instead taking the 50 %.

For the second concern: Sorry that was the editorial error. The design effect is 2.

For the third concern: We have inserted a formula and explanation.

For the fourth concern: the sampling interval was 12. We have written it (page 6; lines 129-131) 

For the final concern: a schematic presentation is attached as Fig1.

Operational Definition

The authors' choice of this instrument over the Multidimensional Scale of Perceived Social Support, which has superior psychometric qualities, remains unclear. I require compelling justification. 

Response: We tried to provide explanation under the general comment 2

Operationalize what is “older people”, “Young – old”, “Middle – old”, and “Old-old” mean and use appropriate references.

Response: We have operationalized the terms with appropriate references (Page 10; lines 193-197). 

Substance addiction is listed as an explanatory variable, which requires operationalization using appropriate tools. Define the terms "ever substance users" and "current substance users." Refer to ASSIST tools.

Results: Thank you for the concern. Yes, you are right. While we see the ASSIST tool, it is a better alternative to operationalize the substance use. However, in the current study, substance use (alcohol, cigarette, and khat) were operationalized using other WHO references (Page 11; lines 213-220). Hope we will consider for our future research since we already missed it.

Of the 830 participants, 816 were enrolled in the study. Explain why 14 were excluded. If there is missing ,explain way of management.

Response: Even if the data collection were supervised daily by assigned supervisors, among 14 responses some were incomplete records and few of them were lost during handover of the records. 

What is your benchmark for categorizing variables such as money, living conditions, and family size? 

Response: Money is categorized by conducting a principal component analysis using a wealth index questions and the categories are based on the EDHS classification. The living condition and family size were classified considering the possible living conditions and family size of the general population in Ethiopia, respectively. 

Table 1 has contradictory data, such as "Having children /live/…. Yes 764 (93.6)" and "Number of live children (n =766)".Please clarify whether this is correct or incorrect.

Response: Thank you for your great observation. It is the editorial error. As it is written in the regression table, the correct one is 766. 

MAGNITUDE PREVALENCE OF PERCEIVED SOCIAL SUPPORT

 Use the word prevalence instead of magnitude

Response: We have done it. 

 When reporting the prevalence of perceived social support among older individuals, it is preferable to provide simply the prevalence (percentage) of low perceived social support with a 95% confidence interval because the prevalence of high perceived social support among these populations is already known.Because your stated low PSS.

Response: We agreed and write only the prevalence of low perceived social support with 95% confidence interval. 

HEALTH CONDITION OF THE STUDY PARTICIPANTS

 Please explain whether the chronic medical and mental diseases listed in Table 2 were diagnosed by a physician, as well as what specific illnesses included in the chronic medical illness category.

Response: The presence of chronic diseases were assessed by asking participants whether they have any chronic disease diagnosed by physicians or having follow-up and taken medication for the disease. To make it clear we have written this under the operational definition (Page 9; lines 186-188). The likelihood of having mental disease was assessed by using K10 (Kessler psychological distress scale).

NUTRITIONAL, BEHAVIORAL, AND PSYCHOSOCIAL CHARACTERISTICS OF THE STUDY PARTICI

---

## [Decision Letter · Decision Letter 1]

4 Nov 2024

Perceived social support and associated factors among older people living in metropolitan cities of northwest Ethiopia: A community-based cross-sectional study

PONE-D-24-24802R1

Dear Dr. Mekonnen,

We’re pleased to inform you that your manuscript has been judged scientifically suitable for publication and will be formally accepted for publication once it meets all outstanding technical requirements.

Kind regards,

Wudneh Simegn

Academic Editor

PLOS ONE

Reviewers' comments:

Reviewer's Responses to Questions

**Comments to the Author**

1. If the authors have adequately addressed your comments raised in a previous round of review and you feel that this manuscript is now acceptable for publication, you may indicate that here to bypass the “Comments to the Author” section, enter your conflict of interest statement in the “Confidential to Editor” section, and submit your "Accept" recommendation.

Reviewer #1: All comments have been addressed

Reviewer #2: All comments have been addressed

2. Is the manuscript technically sound, and do the data support the conclusions?

Reviewer #1: Yes

Reviewer #2: Yes

3. Has the statistical analysis been performed appropriately and rigorously? 

Reviewer #1: Yes

Reviewer #2: I Don't Know

4. Have the authors made all data underlying the findings in their manuscript fully available?

Reviewer #1: (No Response)

Reviewer #2: Yes

5. Is the manuscript presented in an intelligible fashion and written in standard English?

Reviewer #1: Yes

Reviewer #2: Yes

6. Review Comments to the Author

Reviewer #1: (No Response)

Reviewer #2: (No Response)

7. PLOS authors have the option to publish the peer review history of their article (what does this mean?). If published, this will include your full peer review and any attached files.

Reviewer #1: No

Reviewer #2: **Yes: **Agmas Wassie Abate

---

## [Editor Report · Acceptance letter]

13 Nov 2024

PONE-D-24-24802R1 

PLOS ONE

Dear Dr. Mekonnen, 

I'm pleased to inform you that your manuscript has been deemed suitable for publication in PLOS ONE. Congratulations! Your manuscript is now being handed over to our production team.

Kind regards, 

on behalf of

Dr. Wudneh Simegn 

Academic Editor

PLOS ONE